# Current Promising Biomarkers and Methods in the Diagnostics of Antiphospholipid Syndrome: A Review

**DOI:** 10.3390/biomedicines9020166

**Published:** 2021-02-08

**Authors:** Pavla Bradacova, Ludek Slavik, Jana Ulehlova, Adela Skoumalova, Jana Ullrychova, Jana Prochazkova, Antonin Hlusi, Gayane Manukyan, Eva Kriegova

**Affiliations:** 1Masaryk Hospital Usti nad Labem, Department Clinical Hematology, 40113 Usti nad Labem, Czech Republic; pavla.bradacova@kzcr.eu (P.B.); jana.ullrychova@kzcr.eu (J.U.); 2Department of Hematology-Oncology, Faculty of Medicine and Dentistry, University Hospital Olomouc, Palacky University Olomouc, 77900 Olomouc, Czech Republic; jana.ulehlova@fnol.cz (J.U.); jana.prochazkova@fnol.cz (J.P.); antonin.hlusi@fnol.cz (A.H.); 3Department of Internal Medicine III-Nephrology, Rheumatology and Endocrinology, University Hospital Olomouc and Faculty of Medicine and Dentistry, Palacky University Olomouc, 77900 Olomouc, Czech Republic; adela.skoumalova@fnol.cz; 4Department of Immunology, Faculty of Medicine and Dentistry, Palacky University Olomouc and Faculty Hospital, 775 15 Olomouc, Czech Republic; martindihel@seznam.cz (G.M.); eva.kriegova@fnol.cz (E.K.)

**Keywords:** antiphospholipid syndrome, thrombosis, seronegative APS, lupus anticoagulant, anti-cardiolipin, anti-β2-glycoprotein-I, anti-phosphatidylserine/prothrombin, anti-cardiolipin/vimentin, anti-annexin, ELISA, chemiluminescence analysis, multiplex fluorescence flow immunoassay, fluorescence enzyme immunoassay, line immunoassay

## Abstract

Antiphospholipid syndrome (APS) is a hypercoagulation condition associated with the incidence of heterogenic antiphospholipid antibodies (aPLs), which non-specifically affect hemostasis processes. APS is clinically manifested by recurrent arterial and venous thromboses and reproduction losses. The aPL antibodies, which may induce clinical manifestations of APS, include criteria antibodies anti-cardiolipin, anti-β2-glycoprotein-I, and lupus anticoagulant, but also non-criteria antibodies, for example anti-β2-glycoprotein-I domain I, anti-phosphatidylserine/prothrombin, anti-annexin V, and many others. APS occurs mostly in patients of younger and middle age, most frequently in females. Laboratory diagnostics of APS are quite difficult, as they include a wide spectrum of examining methods, which are based on various principles of detection and are performed using various laboratory techniques. The objective of the review is to describe the current state of potentially examined biomarkers and methods in APS diagnostics. The aforementioned biomarkers are lupus anticoagulant, anti-β2-glycoprotein-I, anti-cardiolipin, anti-β2-glycoprotein-I domain I, anti-phosphatidylserine/prothrombin, anti-β2-glycoprotein-I IgA, anti-cardiolipin IgA, anti-annexin V and II, anti-prothrombin, anti-cardiolipin/vimentin, anti-protein S/protein C, and antibodies against phospholipid antigens for whose diagnostics we may use some of the methods established for a long time and some of the modern methods—the coagulation method for the determination of lupus anticoagulant (LA), enzyme-linked imunosorbent assay (ELISA), chemiluminescence analysis (CLIA), multiplex fluorescence flow immunoassay (MFFIA), fluorescence enzyme immunoassay (EliA), line immunoassay (LIA), multiline dot assay (MLDA), and thin-layer chromatography (TLC). Conclusion: Antibodies against phosphatidylethanolamine, phosphatidic acid, phosphatidylserine, phosphatidylinositol, cardiolipin/vimentin complex, and annexin V are currently the most studied new markers. However, these assays have not been standardized until now, both from the laboratory and clinical point of view. In this review we summarize the evidence of the most studied aPL markers and their potential clinical significance in seronegative APS (SN-APS).

## 1. Introduction

Antiphospholipid syndrome (APS), also known as Hughes syndrome, was reported for the first time in 1983 by Dr. Graham Hughes [1]. APS is an autoimmune disease associated with persistent antiphospholipid antibodies (aPLs). The main target of the aPLs is binding to the phospholipid membranes of platelets with their subsequent activation. However, they also bind to endothelia, monocytes, and neutrophils with a procoagulation effect [2,3]. Antiphospholipid antibodies also interfere with the activation of the complement. All this may subsequently result in the development of thrombosis [4]. APS may be primary and also secondary. Primary APS is a condition in which the patient has no other autoimmune disease. Secondary APS occurs in relation with another autoimmune disease: systemic lupus erythematosus (SLE) [5,6,7].

Prevalence of aPLs in the population is approximately 1–5%, but only a minor part develops APS [8]. However, APS is considered to be the most common cause of acquired thrombophilia despite this fact. Clinical manifestations of APS are very variable. Venous thromboses may be manifested by phlebothrombosis of the lower or upper limbs, or by pulmonary embolism. Myocardial infarction or cerebrovascular accident is usually a consequence of arterial thromboses. In the group of pregnancy-related complications, APS is frequently a cause of preeclampsia, miscarriages, premature labor, growth retardation of the fetus due to an insufficient placenta, or death of the fetus. Migraine, immune thrombocytopenia, transient ischemic attack, livedo reticularis, autoimmune hemolytic anemia, and many others were observed as other non-criteria clinical manifestations of APS [9]. Progression of catastrophic antiphospholipid syndrome (CAPS) occurs in approximately 1% of patients with APS, whereby the patient is affected by thromboses mostly in small vessels, leading to multiorgan failure. CAPS is a very severe condition with high mortality [10,11].

Criteria for APS according to the Sydney classification are very strictly defined; at least one clinical and at least one laboratory criterion must be met. Clinical criteria of APS include the occurrence of arterial or venous thromboses and reproduction losses [12,13]. Up to 10–20% of recurrent reproduction losses and up to 20% of cerebrovascular accidents in patients below the age of 50 are caused by APS [14,15]. Laboratory criteria include positivity of at least one antibody of the anti-cardiolipin (aCL) IgG and IgM, anti-β2-glycoprotein-I (anti-β2GPI) IgG and IgM, and the lupus anticoagulant (LA) type [16]. In order to meet the laboratory criteria, the aPLs must be repeatedly positive in an interval of 12 weeks [17]. It is evaluated whether this is single, double, or triple positivity, since patients with triple positivity have the highest risk of thromboses and recurrent miscarriages [18,19,20]. It is required to avoid laboratory examination of APS during ongoing infection due to false positivity of the aPLs [21].

## 2. Antiphospholipid Antibodies

There is a wide range of antiphospholipid antibodies that interact with negatively charged phospholipid surfaces of many cells and tissues by various mechanisms. These aPLs, described on Figure 1, include APS criteria antibodies of the lupus anticoagulant, anti-cardiolipin, anti-β2-glycoprotein-I type, and APS non-criteria antibodies of the anti-β2-glycoprotein-I domain I (anti-DI), anti-annexin V, anti-annexin II, anti-prothrombin (anti-PT), anti-phosphatidylserine/prothrombin (anti-PS/PT), anti-cardiolipin/vimentin (aCL/Vim), anti-protein S/protein C (anti-PS/PC) type, and others.

### 2.1. APS Criteria Antibodies

#### 2.1.1. Lupus Anticoagulant

Lupus anticoagulants are a heterogenic group of immunoglobulins that specifically aim at epitopes of negatively charged protein binding phospholipids of the cellular membrane, prothrombin, and beta2-glycoprotein I, which in vitro prolongs the coagulation tests dependent on phospholipids when there is competition with coagulation factors for phospholipids [22]. 

Positivity of LA is a much more risky factor for the development of thromboembolism, cerebral ischemia, and recurrent reproduction losses in comparison with aCL and anti-β2GPI and even other non-criteria antibodies [23]. LA was demonstrated in 69% in a group of 192 patients with APS [24]. Choi et al. [25] carried out a retrospective study of 833 patients with a persistent presence of aPLs and they found that 46.9% of 96 patients with clinical manifestations of APS had positive LA vs. a group of 737 asymptomatic carriers, where the incidence of LA was only 25.6%. There were no significant differences between the two groups in other aPLs.

#### 2.1.2. Anti-β2-Glycoprotein-I 

β2-glycoprotein-I is anionic glycoprotein with five domains binding to phospholipids. Four domains have regular, conserved sequences, but the fifth domain is aberrant. This domain contains of the insertion of six residues, C-terminal extension of 19 residues, and another disulphide bond that includes the C-terminal cysteine. These additional amino acids in domain V are responsible for unique characteristics of this CUP domain because they form a large positively charged patch that determines affinity to anionic phospholipids [26]. 

The anti-β2GPI IgG and IgM antibody plays a major role in the pathogenesis of APS. Its presence is very strongly associated with thromboembolic complications. The β2-glycoprotein-I molecule consists of five homologous domains and occurs in two conformations, either in a closed circular form or in an open form. In the circular form, there is interaction with anti-β2GPI mainly between domains 1 and 5; in the open form, epitope is uncovered on domain 1, to which anti-β2GPI binds. 

Detection of anti-β2GPI IgG (Figure 2) and IgM (Figure 3) is performed by the enzyme-linked imunosorbent assay (ELISA) method according to the international guideline of the Society of Thrombosis and Haemostasis Scientific and Standardization Committee ISTH SSC. The determined cut-off (99th percentile) in the enzyme-linked imunosorbent assay (ELISA) for positivity of anti-β2GPI is >40 IgG antiphospholipid units/mL (GPL), or IgM antiphospholipid units/mL (MPL) [16]. According to Liu et al., anti-β2GPI IgG is the best predictor of arterial thrombosis, with an odds ratio (OR) = 6.5 [24]. Demonstration of anti-β2GPI IgG has higher specificity for APS than aCL IgG, but lower sensitivity for APS than demonstration of aCL IgG at the same time [27]. However, the results of anti-β2GPI do not always significantly correlate with clinical manifestations of APS, which may be due to insufficient standardization of the ELISA method [28,29,30,31,32]. The modern method of anti-β2GPI detection is the chemiluminescence analysis (CLIA), in which the cut-off for positivity is >20 chemiluminescence unit (CU) (99th percentile) [33]. Multiline dot assay (MLDA) is also an available method. 

#### 2.1.3. Anti-Cardiolipin

Anti-cardiolipin antibodies include a group of antibodies against the cardiolipin part of the VDRL (venereal disease research laboratory) antigen, which are the antibodies that react with phospholipids of the prothrombin activator complex and antibodies that can react with cardiolipin in the fixed phase [34].

aCL IgG is much more associated with cerebral thromboses and myocardial infarctions than aCL IgM. Detection of aCL may be performed by ELISA, CLIA, and MLDA. The determined cut-off (99th percentile) in ELISA for positivity of aCL is >40 GPL/MPL [16]. The cut-off recommended by the manufacturer in CLIA for positivity of aCL is >20 CU (99th percentile) [24].

### 2.2. APS Non-Criteria Antibodies

#### 2.2.1. Anti-β2-Glycoprotein-I Domain I

The presence of APS anti-DI antibodies correlates more significantly with the incidence of thromboses and reproduction losses against other aPLs [35]. The occurrence of anti-DI together with LA is significantly associated with patients with APS and venous thrombosis [27]. Sensitivity of anti-DI after APS of 85% and specificity of 99.5% point to quite great usefulness of anti-DI for APS diagnostics, however, more studies are still needed [36]. Radin et al. [37] analyzed 11 studies involving 1218 patients with APS, where positivity of anti-DI was demonstrated in 45.4%. Tonello et al. [38] carried out a study of 105 patients with APS and persistent presence of the aPL criteria and they demonstrated anti-DI in 41.9%. Positivity of anti-DI was significantly associated with triple positivity. On the contrary, anti-DI negativity was significant in patients with an isolated presence of other aPL criteria. The cut-off recommended by the manufacturer for positivity of anti-DI in CLIA is >20 CU (99th percentile) [24,33]. Serrano et al. specified their own cut-off of >23.8 units (99th percentile) in ELISA for anti-DI in a measurement of 321 healthy volunteers [39]. Slavík et al. [40] examined 74 patients with APS who had positivity at least in one aCL and anti-β2GPI class at the same time. They demonstrated positivity of anti-DI in 21 samples, of which 57% had clinical manifestations of APS. They increased the predictive value for thrombosis from 25% to 68% in anti-DI positive patients by an examination of anti-DI.

#### 2.2.2. Anti-β2-Glycoprotein-I IgA

Antibodies of the IgA class are produced by B-lymphocytes, which may be found in the mucosae, therefore, IgA are also called mucosal antibodies; they are the most common antibodies in the body. IgA antibodies are structurally similar to IgF, but IgA more frequently occur as dimers (Figure 4). The basic function of IgA is to block bacterial adhesion molecules and their opsonization. IgA do not active the complement.

Positivity of the anti-β2GPI IgA class, but with LA negativity at the same time, may be a cause of recurrent unexplainable reproduction losses in females [27,41]. Positivity is put in relation with thrombocytopenia, livedo reticularis, and pulmonary hypertension, and it increases the risk of fatal graft rejection in patients after kidney transplantation [42]. Anti-β2GPI IgA antibodies are more associated with APS than with anti-β2GPI IgM [43]. Ruiz-Garcia et al. performed ELISA measurement of anti-β2GPI IgA in 156 patients with clinical criteria of APS and they demonstrated isolated positivity of anti-β2GPI IgA in 22.4% [44]. Vlagea et al. [45] carried out a study for the presence of anti-β2GPI IgA (cut-off >20 U/mL 99th percentile, 100 healthy follow-ups) in 314 patients with APS and SLE. The presence of isolated positivity of anti-β2GPI IgA in the group of APS was detected only in 7.2%, whereas the presence was detected in 76.2% in the SLE group. Chayoua et al. [46] analyzed a multicentric study of aPL detection in 1068 patients from 8 sites by 4 various methods (CLIA, ELISA, multiplex fluorescence flow immunoassay (MFFIA), fluorescence enzyme immunoassay (EliA)) and they determined isolated positivity of anti-β2GPI IgA in patients with clinical manifestations of APS in 0.3–5% dependent on the device used.

#### 2.2.3. Anti-Cardiolipin IgA

The significance of aCL IgA for the development of thrombotic complications has also been of much interest recently [47]. Using CLIA (cut-off recommended by the manufacturer >20 CU), Liu et al. detected aCL IgA in 192 samples of APS in 42%, in 90 samples of seronegative APS (SN-APS) in 12%, and in healthy donors in 0% [24].

#### 2.2.4. Anti-Prothrombin and Anti-Phosphatidylserine/Prothrombin Complex

The anti-PT IgG antibody may be a very useful predictive factor for the development of thrombosis in patients with SLE [48]. Anti-PT is capable of a bond even to the PS/PT complex. Positivity of anti-PS/PT IgG, IgM with positivity of LA at the same time is very significantly associated with arterial and also venous thromboses and pregnancy complaints [49,50,51] and sensitivity, and specificity for APS is also higher than during positivity of aCL [52]. Using ELISA (cut-off >30 [53]), Liu et al. detected anti-PS/PT IgG, IgM in samples of APS in 72%, in SN-APS in 36%, and in healthy donors in 0%. Anti-PS/PT was more commonly detected in the group of APS and SN-APS than aCL IgG and IgM and anti-β2GPI IgG and IgM. They further found out that particularly anti-PS/PT IgG is the best predictor for deep vein thrombosis, OR = 9.2 [24]. Hui shi et al. found in a study of 186 samples with APS + SN-APS that if LA is positive together with anti-PS/PT, then the OR for the development of thrombosis is 101.6 [54].

#### 2.2.5. Anti-Annexin V and Anti-Annexin II

Annexins are in the group of Ca2^+^-dependent proteins binding phospholipids. Annexin V is the main part of trophoblast and vascular endothelia. Annexin V binds phospholipids with anticoagulation activity; it serves as a so-called protective shield. This shield may be impaired in case of the interaction of annexin V with antibodies, causing thrombosis and reproduction losses [55]. However, the correlation of anti-annexin V with pregnancy complications is not completely significant and more studies are needed [56]. Annexin II is important for the bonding of β2GPI to endothelium and to monocytes. Using the ELISA method, Canas et al. [57] found that demonstration of anti-annexin II is significantly higher in patients with APS than in healthy donors and patients with SLE without thrombosis. However, sensitivity is quite low despite this fact, since anti-annexin II was demonstrated only in 25% of patients with APS.

#### 2.2.6. Anti-Cardiolipin/Vimentin

Vimentin is a part of endothelial cells and may be present even on the surface of apoptotic neutrophils, T-lymphocytes, activated macrophages, and platelets. Vimentin and cardiolipin act on the surface of apoptotic cells as immunogens and may induce the production of antibodies. The presence aCL/Vim is strongly associated with recurrent thrombosis and pregnancy morbidity [52,58]. Ortona et al. demonstrated the presence of aCL/Vim by the ELISA method in patients with APS in 92.5%, in patients with SN-APS in 55.2%, and in patients with SLE in 43.3%. Positivity of aCL/Vim was not demonstrated in any case in a group of healthy donors [59].

#### 2.2.7. Anti-Protein S/Protein C

The mechanism of action of anti-PS/PC is their bond to complexes of phospholipids with coagulation inhibitors protein S and protein C; this results in blocking their activity and subsequently the development of thrombosis. Anti-PS/PC is usually a frequent cause of pregnancy complications and preeclampsia. However, positivity of anti-PC/PS has lower sensitivity and also specificity for APS in comparison with aCL IgG [59].

#### 2.2.8. Antibodies Against Phospholipid Antigens

This group of antiphospholipid antibodies includes antibodies against phosphatidic acid (anti-PA), phosphatidylserine (anti-PS), phosphatidyletanolamine (anti-PE), phosphatidylinositol (anti-PI), phosphatidylcholine (anti-PC), phosphatidylglycerol (anti-PG), lyso-bis-phosphatidic acid (anti-LBPA), and a mixture of phospholipids (APhL). Natural IgG antibodies to the above-mentioned types of lipids are ubiquitously distributed in sera of healthy humans and are believed to serve beneficial functions. Although natural antibodies to lipids generally exhibit germ line or near germ line binding specificities, the antibodies commonly increase transiently in the acute phases of most, if not all, infectious diseases and may serve as a first line of defense [60]. Some studies show that anti-PE may be a cause of fetal loss. Even anti-PS, which inhibits production of choriogonadotropin hormone (HCG), may act similarly [27,61]. Korematsu et al. [62] reported increased levels of anti-PC and anti-PE in three children with cerebral infarction. The anti-LBPA antibodies were demonstrated in a significant number of patients with APS, however, sensitivity and specificity were lower than in aCL and anti-β2GPI [63]. Castanon et al. [64] examined various aPL IgMs and IgGs in 548 serum samples using the ELISA method. Comparison of two groups of APS/healthy donors demonstrated the presence of APhL in 89.7/0%, anti-PI in 89.7/32.1%, anti-PS in 86.2/7.1%, aCL in 93.1/32.1%, and anti-β2GPI in 86.2/0%. Park et al. [65] demonstrated by line immunoassay (LIA) detection that single positivity of anti-PS (OR 16.5) and anti-PA (OR 9.6) is a better predictive factor for thrombosis than positivity of anti-β2GPI (OR 5.5).

## 3. Methods

Table 1 summarizes the available methods for detecting antibodies in the diagnostis of APS based on the principle and technique of the procedure.

### 3.1. Liquid-Phase Assay

#### Lupus Anticoagulant

LA examination should be performed based on the international guideline ISTH SSC for detection of lupus anticoagulant [66]. Detection is based on the ability of present antiphospholipid antibodies in the plasma of the patient to extend the coagulation time in a test dependent on phospholipids. The following basic tests are recommended: dilute Russell’s viper venom time (DRVVT) and activated partial thromboplastin time (aPTT) [67]. The traditional procedure of the LA examination is performed in three basic steps: 1—screening, 2—mixed tests, and 3—confirmation [68]. The LA results are interpreted according to ISTH SSC as positive/negative based on the normalized ratio (NR) calculation (NR = patient/polled normal plasma (PNP)) [69]. With regard to the use of different analyzers and reagents, each laboratory should determine its own cut-off for LA (99th percentile) by measuring ideally 120 (minimally 40) healthy controls [47,70,71]. Cohen et al. [72] carried out a survey in 575 laboratories by means of a “Lupus Program,” the External quality Control of diagnostic Assay and Test Foundation (ECAT). Despite the ISTH SSC guidelines, only 55% of laboratories performed the tests in the screen-mix-confirm order, 50% of laboratories used their own cut-off determined at the 99th percentile, and 46% for interpretation of the results as an NR. Many different laboratories used a “universal” NR >1.2 for interpretation of their DRVVT results. Pradella et al. [73] carried out a DRVVT examination in 200 healthy donors and determined a cut-off NR > 1.22 for positive LA.

### 3.2. Solid-Phase Assay

#### 3.2.1. Enzyme-Linked Imunosorbent Assay

ELISA is the gold standard for detection of many aPLs. The bond of the aPL antibodies in examined plasma/serum to the surface of a microtiter plate hole coated with a fixed phase is the principle of this sandwich method, when a complex antigen/antibody is formed. Human Ig and peroxidase conjugate is bound to this complex. Peroxidase enzyme cleaves a specific chromogenic substrate, producing a color change, the intensity of which is detected through photometry by a reader at a wavelength of 450 nm [74,75,76,77]. The aPL results are obtained by reading the measured optic density from the calibration curve and they are usually indicated in arbitrary units IU/mL or in GPL/MPL units. The cut-off differs for the individual aPLs. Serrano et al. determined a cut-off >20 units in anti-β2GPI IgA using ELISA (99th percentile) by measuring 321 healthy volunteers [39].

The test results of various kits in various laboratories show quite large variability. Due to this reason, the results of aPL tests often do not provide a sufficient benefit for the clinical use; the method needs to be more standardized [28,29,30,31].

#### 3.2.2. Fluorescence Enzyme Immunoassay

The EliA method is based on a similar principle as ELISA, except that the conjugate contains mouse Ig and β-galactosidase. Detection is based on fluorescence intensity, which is optically demonstrated in the detector. The cut-off for positivity of aCL and anti-β2GPI recommended by the manufacturer is >10 U/mL [78]. Bor et al. determined their own cut-off (99th percentile) in 377 samples of patients with APS for the individually determined aPLs and compared this with the cut-off recommended by the manufacturer. They subsequently found that based on their own cut-off they evaluated 40 positive samples fewer than in the cut-off determined by the manufacturer [79].

#### 3.2.3. Chemiluminescence Immunoassay

CLIA is a method of quantitative detection of aCL IgG, IgM, anti-β2GPI IgG, IgM, and anti-DI. CLIA is a very well-standardizable method, performed using an automatic analyzer, and it is suitable for a higher number of samples [80]. The bond of the aPLs in the examined serum/plasma sample to paramagnetic particles coated by an appropriate surface is the principle of CLIA. Isoluminol-labeled compatible human Ig is bound to this formed complex. A chemiluminescence reaction is initiated after the addition of a triggering reagent [81]. The emission of light occurs during the chemiluminescence reaction; this is detected by an optic module in the device in relative light units (RLU). Measured RLU are directly proportional to the concentration of the individual aPLs in the sample. Measured RLU are converted to chemiluminescence units (CU) by means of a logistic curve 4PLC. The cut-off recommended by the manufacturer is >20 U/mL [82]. Chayoua et al. [83] carried out a multicentric study in 1168 samples. They compared the results of aCL IgM and IgG and anti-β2GPI IgG and IgM in three solid-phase assays (MFFIA, EliA, ELISA) and found that the best correlation (0.900) in anti-β2GPI IgG was between MFFIA and CLIA. On the contrary, the worst correlation (0.514) in aCL IgM was between MFFIA and EliA. Salma et al. [84] compared CLIA and ELISA in 370 samples and demonstrated a similar sensitivity of both methods for aCL IgG and IgM and anti-β2GPI IgM, but CLIA had higher sensitivity for anti-β2GPI IgG than ELISA.

#### 3.2.4. Multiplex Flow Fluorescence Immunoassay

MFFIA analysis for the detection of aCL IgG and IgM and anti-β2GPI IgG and IgM is based on the use of paramagnetic particles coated with an appropriate antigen to which the aPLs are bound in the sample. A conjugate of human Ig with fluorescein phycoerythrin is subsequently added. Fluorescence is identified in relative fluorescence units (RFI) as the particles pass through the detector. The method is performed using an automatic analyzer and is suitable for performing on a larger number of samples [85]. The cut-off for positivity of aCL and anti-β2GPI recommended by the manufacturer is >20 U/mL. Grossi et al. [86] compared the results of 134 patients on MFFIA and CLIA and demonstrated a very good compliance between both methods. Compliance for aCL IgG was 88.1%, and for anti-β2GPI IgG was 97.8%.

#### 3.2.5. Multiline Dot Assay

MLDA is a semi-quantitative method for detection of multiple aPLs at the same time, performed on polyvinylidene difluoride (PVDF) membranes. Various immobilized phospholipids are piled up on PVDF in strips, to which the respective aPLs from the serum sample are bound. Detection is performed using densitometry, and the results are indicated as positive/negative [87]. Compared to ELISA, hydrophobic PVDF membranes imitate the bond of the aPLs in vivo, they are more porous, and may hide a large portion of the phospholipid hydrophobic part, which may result in denser expression of the phospholipid hydrophilic part on the PVDF surface and intensified interaction with the examined aPLs. Misasi et al. [27] and Egerer et al. [88] performed a comparative MLDA and ELISA study, and in the measurement results they demonstrated a good to very good compliance of aCL and anti-β2GPI between both methods. Using MLDA, aCL and anti-β2GPI and the presence of other aPLs may be determined in the sample. The method is not certainly suitable for an analysis of a larger number of samples due to characteristics of its implementation, and standardization of MLDA is not completely sufficient either.

#### 3.2.6. Line Immunoassay

LIA is a novel multiline assay for the determination of up to 10 different aPLs at the same time. Various phospholipids are immobilized on the PVDF membrane with no addition of a cofactor, and binding of the aPLs is dependent only on β2GPI present in the examined sample. Ig and peroxidase conjugate cleave the substrate. Individual strips are analyzed qualitatively using positive/negative densitometry. The optical density (oD) cut-off for positivity is ≥50 of oD (determined in 150 healthy donors, 99th percentile). Thaler et al. compared an examination of 10 different aPLs by the LIA method in 53 APS and 34 healthy controls with CLIA and ELISA technologies. The sensitivity of LIA for aCL and anti-β2GPI IgG was significantly higher than in other methods [89]. Roggenbuck et al. [90] and Nalli et al. [91] independently compared the detection of different aPLs by LIA and ELISA in two files of patients with APS and healthy controls and demonstrated a very good compliance between the results in patients with APS. In addition to ELISA, the LIA method could differentiate patients with APS from patients with infectious diseases or asymptomatic carriers probably by exposure of domain I. Park et al. [65] detected 9 different aPLs in 180 patients with APS by LIA and ELISA, and by a comparison of both methods they demonstrated compliance in the results of aCL IgG (68.2%), aCL IgM (82.6%), anti-β2GPI IgG (71.7%), and anti-β2GPI (93.2%). Park et al. demonstrated by LIA detection that single positivity and anti-PS (OR 16.5) and anti-PA (OR 9.6) are better predictive factors for thrombosis than anti-β2GPI (OR 5.5).

#### 3.2.7. Thin-Layer Chromatography TLC

TLC is a non-quantitative screening method performed on phospholipid-coated aluminum plates. TLC is performed in several basic steps: Antigen separation occurs at first, followed by immunostaining with the examined aPLs, and finally immunoreactivity is detected using a chemiluminescence reaction [92,93]. In case of detection of immunoreactivity (positivity) of aPL by TLC, it is subsequently appropriate to perform a targeted examination of the individual aPLs using ELISA. Based on a comparative ELISA and TLC study of 120 samples, Sorice et al. found that TLC shows higher specificity, but lower sensitivity than ELISA [94]. As with MLDA, TLC is not suitable for an analysis of a larger number of samples, and in TLC it is true that there is insufficient standardization of this method.

## 4. Conclusions

### Seronegative APS

In practice, we often find patients with clinical manifestations of APS, but they are repeatedly negative for all of the criteria for an antiphospholipid antibody. They are so-called seronegative APS [52,95,96]. A part of patients with SN-APS show repeated positivity of non-criteria antibodies of the anti-DI, anti-PS/PT IgG and IgM, anti-annexin V IgG and IgM, anti-PS, anti-PA type, and others [58,97]. Trugliia et al. [98] analyzed 61 samples of SN-APS in females with reproduction complications. The aCL antibodies were analyzed using TLC; aCL/Vim antibodies, anti-PS/PT, anti-β2GPI IgA, and aCL IgA were analyzed using the ELISA method. At least one positive aPL was demonstrated in 81.9%. Repeated testing 12 weeks later demonstrated persistent positivity of at least one aPL in 57.4% of females.

Patients with SN-APS are at risk for recurrent thrombotic and pregnancy complications; long-term prophylactic treatment is therefore required [27]. Due to this reason, it has been currently proving increasingly beneficial to revise the original laboratory criteria of APS [16] and to include specifications of other non-criteria antibodies summarized in Table 2 [99]. The introduction of additional aPLs into routine laboratory practice will certainly represent a useful tool for more precise and accelerated APS diagnostics [54,100,101].

## Figures and Tables

**Figure 1 biomedicines-09-00166-f001:**
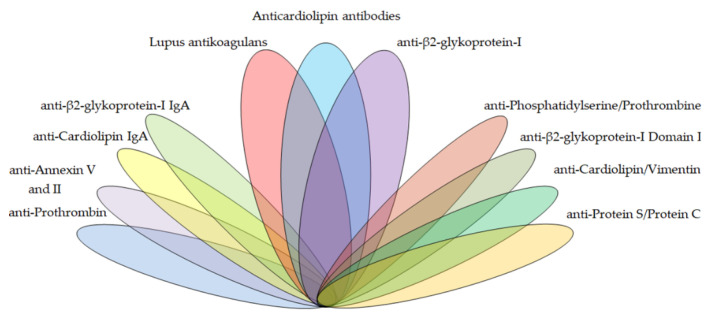
The spectrum potential antiphospholipid antibody targets in the diagnostics of antiphospholipid syndrome (APS).

**Figure 2 biomedicines-09-00166-f002:**
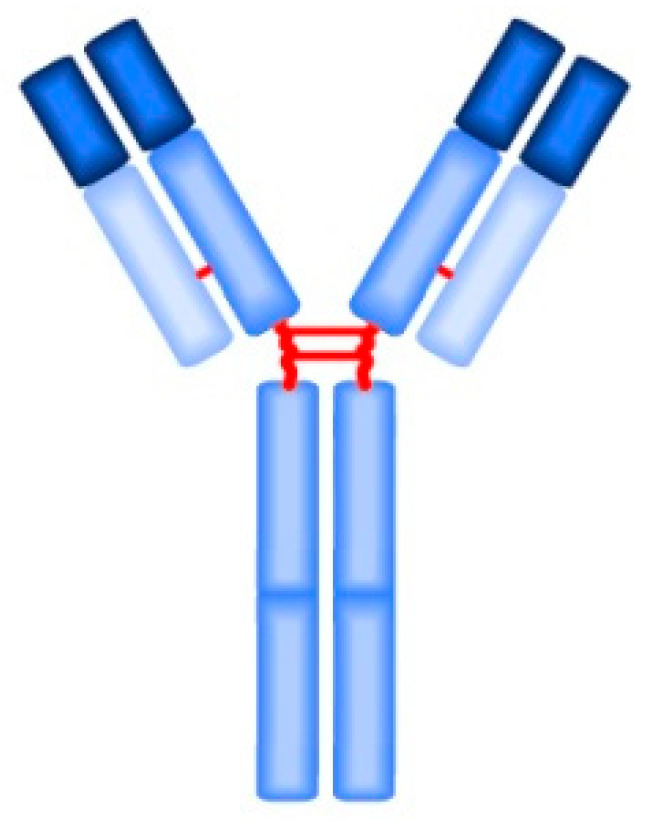
The IgG monomer structure.

**Figure 3 biomedicines-09-00166-f003:**
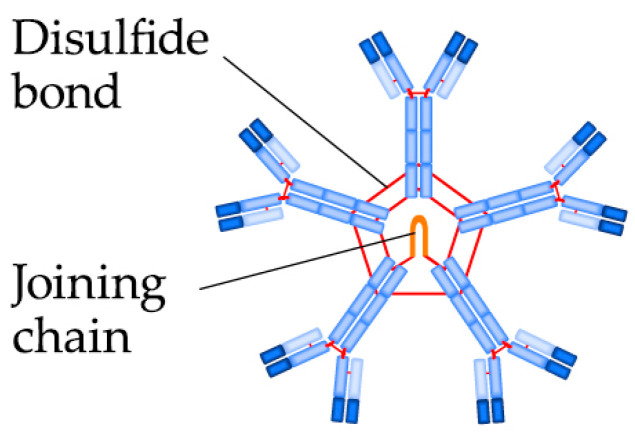
The IgM pentamer structure.

**Figure 4 biomedicines-09-00166-f004:**
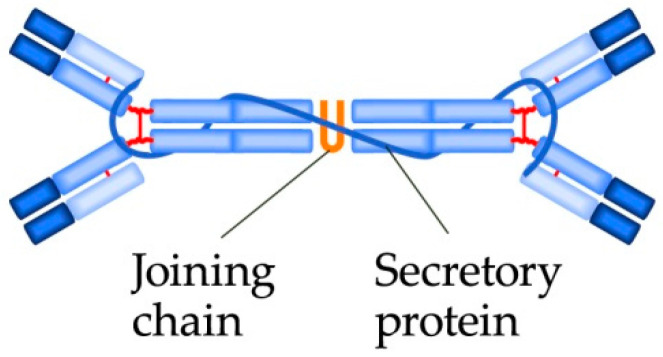
The IgA dimer structure.

**Table 1 biomedicines-09-00166-t001:** Overview of the methods available for the examination of biomarkers.

Methods	Assay	Determination
Dilute Russell’s viper venom time (DRVVT) Activated partial thromboplastin time (aPTT)	Liquid-phase	Quantitative
Enzyme-linked immunosorbent assay (ELISA)	Solid-phase	Quantitative
Fluorescence enzyme immunoassay (EliA)		Quantitative
Chemiluminescence immunoassay (CLIA)		Quantitative
Multiplex flow fluorescence immunoassay (MFFIA)		Quantitative
Multiline dot assay (MLDA)		Semi-quantitative
Line immunoassay (LIA)		Qualitative
Thin-layer chromatography (TLC)		Qualitative

**Table 2 biomedicines-09-00166-t002:** Overview of biomarkers and methods used to study them.

Biomarkers	Methods	Reference
Lupus anticoagulant (LA)	DRVVT aPTT	Liu [24], Choi [25], Pengo [66], Linnemann [68]
Anti-β2-glycoprotein-I (anti-β2GPI) IgG, IgM	ELISA EliA CLIA MFFIA MLDA LIA	Miykis [16], Liu [24], Misasi [27], Serrano [39] Vanouverchelde [78], Bor [79], Chayoua [83] Janek [33], Chayoua [83], Salma [84] Chayoua [83], Chayoua [85], Grossi [86] Misasi [27], Bevers [87], Egerer [88] Park [65], Egerer [88], Thaler [89], Roggenbuck [90], Nalli [91]
Anti-cardiolipin (aCL) IgG, IgM	ELISA EliA CLIA MFFIA MLDA LIA	Miykis [16], Liu [24] Vanouverchelde [78], Bor [79], Chayoua [83] Janek [33], Chayoua [83], Salma [84] Chayoua [83], Chayoua [85], Grossi [86] [Misasi [27], Bevers [87], Egerer [88] Park [65], Egerer [88], Thaler [89], Roggenbuck [90], Nalli [91]
Anti-β2-glycoprotein-I domain I (anti-DI)	ELISA CLIA	Serrano [39] Slavik [40]
Anti-β2-glycoprotein-I IgA	ELISA EliA CLIA MFFIA	Ruiz-Garcia [44], Vlagea [45] Chayoua [46] Chayoua [46] Chayoua [46]
Anti-cardiolipin IgA	CLIA	Liu [24]
Anti-prothrombin (anti-PT) Anti-phosphatidylserine/prothrombin (anti-PS/PT)	ELISA	Liu [24], Shi [54]
Anti-annexin V Anti-annexin II	ELISA	Canas [57]
Anti-cardiolipin/vimentin (aCL/Vim)	ELISA	Ortona [58]
Anti-protein S/protein C (anti-PS/PC)	LIA	Arachchillage [59]
Anti-phosphatidic acid (anti-PA) Anti-phosphatidylserine (anti-PS) Anti-phosphatidyletanolamine (anti-PE) Anti-phosphatidylinositol (anti-PI) Anti-phosphatidylcholine (aPC) Anti-phosphatidylglycerol (aPG) Anti-lyso-bis-phosphatidic acid (anti-LBPA) Anti-mixture of phospholipids (APhL)	ELISA LIA	Castanon [64] Park [65]

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
