# Peer review of "Current Promising Biomarkers and Methods in the Diagnostics of Antiphospholipid Syndrome: A Review"

_biomedicines, 2021, doi:10.3390/biomedicines9020166_

Round 1

Reviewer 1 Report

Bradacova and coauthors reviewed current promising biomarkers and methods in the diagnostics of antiphospholipid syndrome. This ms. is very well written, is clear and concise. In my opinion, the current manuscript meets the quality guidelines to be published in Biomedicines, just with a few minor corrections detailed below.

  1. In figure 2, the structures of other types of antibodies should be included.
  2. A schematic summarizing the different diagnostics methods should be added.
  3. Zwitterionic phosphatidylcholine (PC) is a main type of phospholipid. Authors mentioned that antibodies against phosphatidylethanolamine, phosphatidic acid, phosphatidylserine, phosphatidylinositol, Cardiolipin/Vimentin complex and Annexin V are found, but little information about phosphatidylcholine. Could the authors add some discussion about whether PC induces antibodies?

Author Response

Dear rewiever,

I would like to respond to your comments, which we greatly appreciate. The pictures of the structures have been supplemented according to your requirements and are certainly a clear guide to the formation of antibodies. The discussion of antibody formation was added to Chapter 2.2.8. (red text)

Bets regard

Assoc. Prof. Mgr. LudÄ›k Slavík, PhD.  
Deputy head for laboratory diagnostics

Dept. of Hemato-oncology

University Hospital Olomouc 

I. P. Pavlova 185/6 

779 00 Olomouc

Tel.: +420 588 445 350

web: www.fnol.cz

Reviewer 2 Report

Bradacova et al have summarized the information on the currently available biomarkers for antiphospholipid syndrome (APS) diagnosis with special emphasis on the methods used for their assessment. In addition to the most common antiphospholipid antibodies detected in APS patients, in this review they described the non-criteria antibodies that are detectable in patients considered to be seronegative.

I hope the following comments will help them to improve the manuscript:

- As indicated in the “Instructions for authors” included on the website of the Biomedicines journal: “ Abbreviations should be defined in parentheses the first time they appear in the abstract, main text, and in figure or table captions and used consistently thereafter.” In the main text there are many abbreviations that have been defined in the abstract but not in the text itself. In the main text there are many abbreviations that have been defined in the abstract but not in the text itself. The main text should be carefully reviewed and all abbreviations included in it should be defined for the first time. Some examples are: “aPL” (line 49), “APS” (line 53), “aCL” (line 113, “CLIA”131, “MLDA” line 132, “SN-APS” (line 191)…

- This review provides a broad range of information on biomarkers and methods in APS diagnostics. Since there are many and very varied biomarkers and methods exposed, it would be advisable to include some Tables as a summary. For example, in one column of the table the biomarkers could be included and in another the specific methods used for their study (with the corresponding bibliographic references).

- Line 55:  The sentence wording is confusing if the hyphen before "Systemic lupus erythematosus" is used. It would be advisable to replace the hyphen with a colon.

- Line 69-70. “… At least three organs or tissues are affected; it is 70 frequently multiple organ impairment. “ Review the wording of this sentence as separating it from the previous sentence may make it difficult to understand.

- Line 118: There is a "T" missing at the beginning of the sentence

- Line 118: I would prefer the useich is the bibliographic reference for the information included in this paragraph?

- Figure 2: The text included in the figure must be translated into English

Author Response

Dear reviewer,

we greatly appreciate your feedback, which will improve your handwriting. We all worked out as follows

  1. all abbreviations have been added “aPL” (line 49), “APS” (line 53), “aCL” (line 113, “CLIA”131, “MLDA” line 132, “SN-APS” (line 191)
  2. methods and biomarkers were summarized in two tables, including the authors of the manuscripts
  3. corrected by changing the text separation ....Secondary APS occurs in relation with another autoimmune disease: Systemic lupus erythematosus (SLE) [5-7]. 
  4. corrected to .....Progress of catastrophic antiphospholipid syndrome (CAPS) occurs in approximately 1% of patients with APS, when the patient is affected by thromboses mostly in small vessels, leading to multiorgan failure.
  5. line 118 corrected
  6. Figure 2 was translated

Round 2

Reviewer 1 Report

I recommend it for publication in the current form based on the revision.